# Does Form Follow Function? An Empirical Exploration of the Impact of Deep Neural Network Architecture Design on Hardware-Specific Acceleration

## ABSTRACT

Advances in deep learning during the last decade have led to state-of-the-art performance across a wide variety of tasks. However, one of the biggest challenges with the widespread adoption of deep learning in an operational manner has been high computational complexity. This challenge is particularly important to tackle given the recent proliferation of smart sensors and edge devices. This has led to hardware-specific acceleration platforms designed specifically to accelerate deep neural network inference based on microprocessor architectural traits. While promising, the degree of inference speed gains achieved via hardware-specific software acceleration can vary significantly depending on the design of the deep neural network architecture and the microprocessor being leveraged for inference. The fine-grained relationship between form and function with respect to deep neural network architecture design and hardware-specific acceleration is one area that is not well studied in the research literature, with form often dictated by accuracy as opposed to hardware function. In this study, a comprehensive empirical exploration is conducted to investigate the impact of deep neural network architecture design on the degree of inference speedup that can be achieved via hardware-specific acceleration. More specifically, we empirically study the impact of a variety of commonly used macro-architecture design patterns across different architectural depths through the lens of OpenVINO microprocessor-specific and GPU-specific acceleration. Experimental results showed that while leveraging hardware-specific acceleration achieved an average inference speed-up of 380%, the degree of inference speed-up varied drastically depending on the macro-architecture design pattern, with the greatest speedup achieved on the depthwise bottleneck convolution design pattern at 550%. Furthermore, we conduct an in-depth exploration of the correlation between FLOPs requirement, level 3 cache efficacy, and network latency with increasing architectural depth and width. Finally, we analyze the inference time reductions using hardware-specific acceleration when compared to native deep learning frameworks across a wide variety of hand-crafted deep convolutional neural network architecture designs as well as ones found via neural architecture search strategies. We found that the DARTS-derived architecture to benefit from the greatest improvement from hardware-specific software acceleration (1200%) while the depthwise bottleneck convolution-based MobileNet-V2 to have the lowest overall inference time of around 2.4 ms. These findings illustrate the importance of tailoring network architecture design to account for the intricacies of microprocessor architecture traits to enable greater hardware-specific acceleration.

## CCS CONCEPTS

• **Computing methodologies** → **Machine learning**; • **Computer systems organization** → **Embedded software**.

## KEYWORDS

Neural Networks, Edge Devices, Hardware Acceleration, OpenVINO, Macro-Architectures, Embedded Machine Learning, Design Patterns

### ACM Reference Format:
Anonymous Author(s). 2018. Does Form Follow Function? An Empirical Exploration of the Impact of Deep Neural Network Architecture Design on Hardware-Specific Acceleration. In *Proceedings of TinyML Research Symposium (TinyML '21)*. ACM, New York, NY, USA, 8 pages. https://doi.org/10.1145/1122445.1122456

## 1 INTRODUCTION

The past decade has witnessed significant advances in deep learning [18, 19], with deep neural networks matching or even exceeding human performance in a wide variety of areas such as image classification [12, 14, 17], object detection [8, 21, 26], and image segmentation [7, 10]. Until recently, deep neural networks were typically designed with accuracy as the overriding metric rather than hardware efficiency. This design principle has led to increasingly complex models that require large computational resources to function [1, 3, 27–29]. The increasing complexity of modern deep neural networks poses significant operational challenges for widespread deployment on small edge devices used in different applications such as autonomous vehicles and IoT consumer devices where computational resources are limited. These challenges significantly hinder the applications of deep learning and become particularly important to tackle as edge devices continue to proliferate.

To mitigate the complexity of neural networks, a variety of macro-architecture design patterns have been introduced in recent years, which aim to reduce computational complexity for efficient inference [12, 13, 15, 23, 25, 31]. For example, grouped convolutions have been employed by ResNext [29] to reduce the number of computations required compared to a standard convolution. The premise behind grouped convolutions is to have a filter operating

on only a smaller group of input channels to significantly reduce the number of convolutions that need to be performed. An extreme version of grouped convolution, depthwise separable convolution, has been used with greaxft efficacy with MobileNet [13]. Depthwise separable convolution are similar to grouped convolutions, with the major difference being that the number of groups equaling the number of input channels. Bottleneck convolutions are another example of efficient convolutions and have been used successfully to design neural networks such as DARTS [20], ResNet architectures [12] and Squeezenets [15], where pointwise convolutions are leveraged to decrease and increase channel dimensionality before and after a spatial convolution, respectively. Depthwise convolution and bottleneck convolution can be combined for a further improvement in network efficiency and have been used in MobileNet-V2 for a drastic reduction in inference time [25]. Convolutions can also be factorized to lower the amount of multiplication-accumulation counts required [13, 25]. For example, a $3 \times 3$ kernel can be broken into a $1 \times 3$ and $3 \times 1$ convolutions. However, this presumes that the original kernel can be factorized which may not always be possible.

Another key development towards operational deep learning deployment has been that of hardware-specific acceleration platforms [2, 9, 11, 22, 30]. Such acceleration platforms take advantage of the underlying hardware's architectural traits to speed up neural network inference. For example, an acceleration platform can utilize a microprocessor's vector processing unit or optimize the use of processor cache for optimal network execution. While hardware-specific acceleration is a promising development for deploying deep neural networks on edge devices, the fine-grained relationship between form and function with respect to network architecture design and hardware-specific acceleration is an area that has not been widely studied in the literature.

Motivated to gain a better understanding between form and function from this perspective, we conduct a comprehensive empirical exploration on the impact of deep neural network architecture design on hardware-specific acceleration. Specifically, we investigate the impact of a variety of commonly used macro-architecture design patterns across different architectural depths and convolutional widths on hardware-specific acceleration through the lens of OpenVINO, a software platform designed specifically for hardware-software acceleration of deep neural networks. Furthermore, we examine the impact of level 3 (L3) cache optimization on network latency as well as investigate the correlation between FLOPs requirement, network latency, and L3 cache efficacy. Finally, we study the inference time reductions using hardware-specific acceleration on a variety of hand-crafted deep neural network designs as well as ones found via neural architecture search (NAS) techniques. Finally, based on the findings of this empirical exploration, we present practical considerations in designing efficient deep neural networks tailored for the intricacies of CPU and GPU architecture traits to enable greater hardware-specific acceleration.

## 2 EMPIRICAL EXPLORATION

This study provides a fine-grained empirical exploration on the impact of network architecture design on hardware-specific acceleration. Specifically, this study considers the impact of microprocessor-specific acceleration with an Intel Core i5-7600K, as well as GPU-specific acceleration with an integrated GPU (Intel HD Graphics 630).

This study is comprised of three experiments. In the first experiment explained in Section 2.1, we measure the impact of six different macro-architecture design patterns on hardware-specific acceleration. We conduct this experiment in a parametric fashion by studying the trend of inference speedup gained through hardware-specific acceleration under different architectural depth scenarios. Additionally, this experiment also considers the impact of hardware acceleration under different network widths. Studying the impact of hardware acceleration under different architectural depths is important as we hypothesize that, unlike theoretical complexity analysis using metrics such as the number of floating point operations (FLOPs), hardware-specific inference speedups will likely not follow a simple trend for all design patterns as the depth increases. Similarly, the impact of network width on hardware acceleration is also important to investigate as the efficiency of different macro-architectures may vary with the width of the network architecture. For instance, some design patterns may be more computationally efficient with a wider width and as such would be preferred choices in the deeper layers of a deep convolutional neural network where the channel widths are typically wider.

Building upon the first experiment, we subsequently investigate the relationship between network execution time, FLOPs requirement, and L3 cache access, and the impact of hardware-specific acceleration in Section 2.2. Specifically, we measure the number of times the microprocessor does not find data in the L3 cache and has to fetch data from the much slower DRAM. The primary purpose of this experiment is to gain a deeper understanding on how hardware-acceleration reduces the network latency. Similar to the first experiment, we construct several neural networks with architectural depths ranging from 10 to 50 layers and measure the amount of times the L3 cache does not hold the required data. Similarly, we also increase the architectural width of the neural networks and measure the efficacy of the L3 cache access with and without hardware-specific acceleration. Finally, we correlate the measured L3 miss rate from these experiments with the network execution time and FLOPs requirement of each design pattern investigated in this study.

Finally, the third experiment reported in Section 2.3, measures the inference speedups gained through hardware-specific acceleration on a wide variety of hand-crafted deep neural networks as well as ones found via NAS approaches. Many of these deep neural networks employ the macro-architecture design patterns studied in the first experiment, and thus will give us additional insights under a more complex scenario where additional architecture components are in place to form the well-defined network architecture.

All three experiments are conducted on an Intel Core i5-7600K microprocessor operating at 3.8 GHz as well as an integrated Intel HD Graphics 630 GPU. These devices are leveraged via the

hardware-specific acceleration platform, OpenVINO. The Core i5-7600K processor features Advanced Vector Extensions 2 (AVX2) which enables hardware-specific acceleration platforms to optimize or 'vectorize' network execution. AVX allows for 'single-instruction multiple data' (SIMD) operations where multiple data can be loaded onto the physical registers and operated on with a single instruction, resulting in much faster execution.

The PyTorch [24] microprocessor-specific accelerated models were executed with 32-bit floating point precision. Experimental results showed that quantizing the microprocessor-specific accelerated models to 16-bit did not provide any speed up on the CPU. This is due to the fact that the Intel Core i5-7600K processor promotes 16-bit floats to 32-bit floats [5]. Nevertheless, the integrated GPU (iGPU) does support 16-bit floating point operations and as such the iGPU accelerated models were first quantized to half floats prior to deployment [4].

## 2.1 Form vs. Function: Design Pattern Choices on Hardware-specific Acceleration

The first experiment of this study examines the impact of network architecture design choices on the inference speedup gains through hardware-specific acceleration. For this experiment, we construct a large number of deep neural network architectures using a wide variety of commonly-used macro-architecture design patterns, with architectural depths ranging from 10 to 50 layers. The six macro-architecture design patterns studied in this experiment include: **i)** standard spatial convolution, **ii)** factorized convolution, **iii)** bottleneck convolution, **iv)** depthwise separable convolution, **v)** depthwise bottleneck convolution, and **vi)** grouped convolution. The execution time is measured for each deep neural network architecture against the native deep learning framework (i.e., PyTorch), microprocessor-specific acceleration, and GPU-specific acceleration using OpenVINO. The network execution time is measured with increasing batch sizes to investigate the impact of batch size on hardware-specific acceleration.

As seen in Figure 1, hardware-specific acceleration universally improves network inference speed across all evaluated deep neural network architectures with different macro-architecture design patterns and different architectural depths. Although the greatest gains are provided by the GPU-specific acceleration, the microprocessor-specific acceleration also provides a significant speed up over the native framework. This is made very apparent by the significant speed up of hardware-specific acceleration deployment and the gap between the inference time of the native framework and hardware-specific acceleration as architectural depth increases in all cases. The microprocessor-specific acceleration is particularly promising as many embedded devices do not have a dedicated GPU available. More interestingly, the speed gains from hardware-specific acceleration improve as the network complexity increases for all evaluated macro-architecture design patterns.

Furthermore, it can be observed that the degree of inference speed gains through hardware-specific acceleration varies significantly depending on the macro-architecture design pattern used, with each exhibiting a different speedup trend. More specifically, deep neural network architectures leveraging the depthwise separable convolution design pattern demonstrated the greatest level of

speedup from hardware-specific acceleration; this speed up with the execution time of a 50-layer network architecture under hardware-specific acceleration is at the same level as a much shallower, 10-layer network, architecture under the native framework. Moreover, the speed gains provided by the hardware-specific acceleration generally increases with the batch size. However, the native framework generally slows down as the batch sizes increase. An example of this is with standard convolutions, where the per-image inference time increases from 300 ms to 500 ms as the batch size is increased from 1 to 32 with the native framework. In contrast, the hardware accelerated neural networks provide similar per-image inference time for different batch sizes, resulting in greater and greater speed gains as the batch size increases.

The impact of network width on network execution latency is investigated by constructing several neural network architectures with different network widths ranging from 32 to 512. Similar to the architectural depth study, these networks are implemented with aforementioned commonly used design patterns. As seen in Figure 2, all macro-architectures again benefit from hardware-specific acceleration. In general, all design patterns perform similarly with a narrow channel width. Bottleneck convolution scales poorly with increasing network width, posting even worse numbers than standard convolution. However, standard convolution does not improve significantly with microprocessor-specific acceleration and required GPU-specific acceleration to scale well. Standard convolution, factorized convolution and bottleneck convolution scale with roughly the square root of the channel width. In contrast, depthwise separable convolution, depthwise bottleneck convolution and grouped convolution stay almost linear, resulting in significantly more efficient network execution. Indeed, the microprocessor accelerated model variants perform on par with GPU-accelerated models. This is especially promising for network architectures designed for deployment on embedded systems where a dedicated GPU may not be available. Importantly, these design patterns should thus be the preferred choices for the deeper and wider layers of a deep convolution neural network.

These speed gain variations observed in Figure 1 and Figure 2 are primarily due to the way OpenVINO performs software and hardware-specific acceleration. More specifically, OpenVINO first fuses common operations together to streamline operations. For example, ReLU operations are typically fused with the preceding convolution layers and executed as a single operation. OpenVINO applies the ReLU activation while the convolution operation is already in cache memory, thereby preventing a performance hit by an avoidable access of DDR memory; and as such, the network execution is optimized by taking advantage of efficiently using the underlying memory hierarchy. In the case of the microprocessor, OpenVINO optimizes the execution of all layers by running them via the on-board vector processing unit (AVX2 on the Intel Core i5-7600K).

## 2.2 The Impact of Hardware-Specific Acceleration on Cache Access Efficacy

In this section, we investigate the relationship between FLOPs requirement, network latency, and L3 cache efficacy, and the impact of hardware-specific L3 cache optimization. As seen in Figure 1,

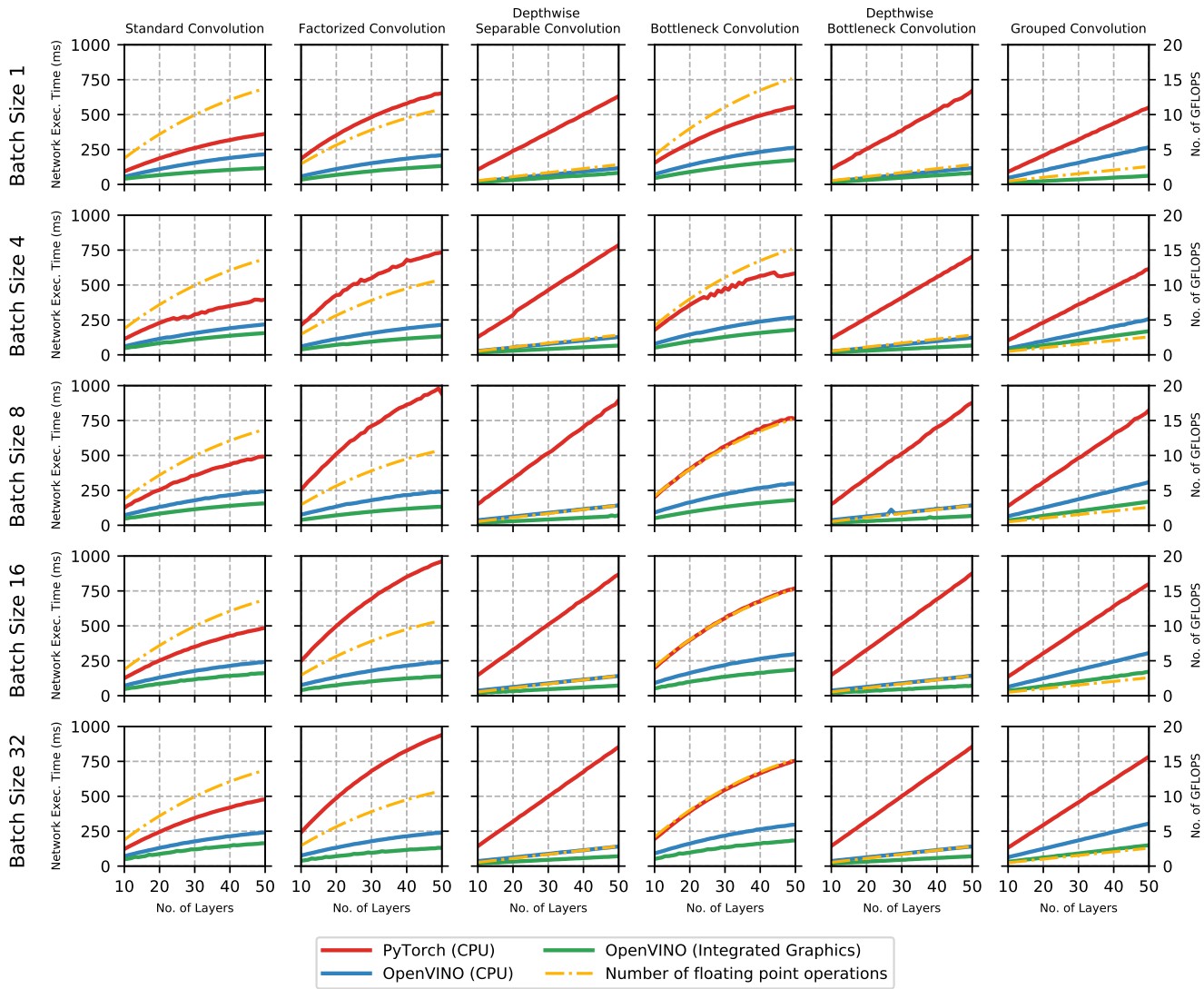

**Figure 1: The impact of macro-architecture design choices on hardware-specific acceleration across different architectural depths. Notably, speed gains are seen from hardware-specific acceleration in all cases. More interestingly, the speed gains from hardware-specific acceleration increases as the network complexity increases in all cases. This is observed by noting that the inference time gap between the native framework and the hardware-specific acceleration increases significantly as architectural depth increases. Furthermore, it can be observed that the depthwise bottleneck convolution design pattern is accelerated the most out of the six macro-architecture design patterns studied here, yielding an improvement of about 550% with a 50-layer network. Finally, it is observed that FLOPs requirement does not necessarily correlate with the network execution time. For example, depthwise separable convolution has one of the lowest FLOPs requirements yet reports one of the highest inference times with native framework. This implies that the speed of computation is being limited in such cases by software implementation. All plots share the same vertical and horizontal scale to allow for a quick visual comparison across the different macro-architectures as well as batch sizes.**

the FLOPs requirement for a design pattern does not necessarily predict the network execution speed. For example, depthwise bottleneck convolution requires the lowest amount of multiplication-accumulation operations yet it reports one of the highest latency in Figure 1 with native frameworks. To assess the impact of optimized cache access, we construct several neural networks with architectural depth of 10 to 50 layers. For each neural network architecture, we measure the number of times the microprocessor was not able to find data in the L3 cache and had to resort to fetching data from the significantly slower DRAM. Figure 3 shows the number of L3

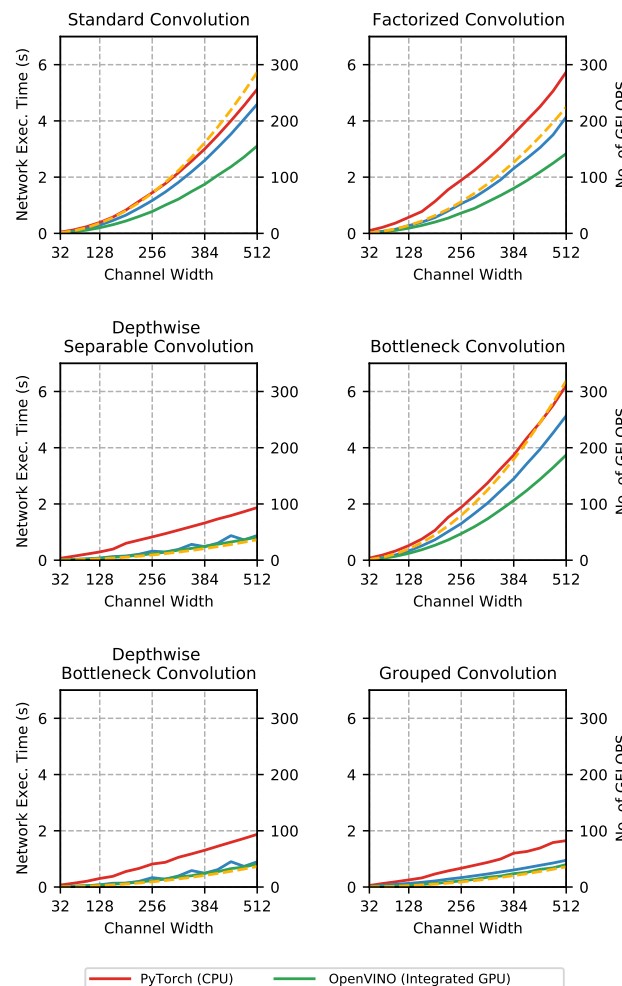

**Figure 2: The impact of hardware-specific acceleration on network inference speed across different network widths. In particular, hardware-acceleration improves network execution across all convolutional widths with all six design patterns. We find that in the case of increasing convolutional widths, the network execution time is correlated with the number of floating point operations for each design pattern. This implies that the number of FLOPS quickly becomes the primary limitation for network execution speed as the number of output channels increases.**

cache miss events, as well as the network latency against the architectural depth for all design patterns explored in this study. The dashed lines denote the number of times the CPU did not find data in the L3 cache. We can note that hardware-specific acceleration improves L3 cache efficacy in all cases to a varying degree. In the case of depthwise bottleneck convolution, the number of missed L3 cache events drops by about an order of magnitude. The improved L3 cache unleashes the true potential of depthwise bottle convolution, improving the network execution by approximately 520%. Similar improvements are noted for depthwise separable convolution. In contrast, for standard convolution the L3 miss rate is also reduced though not to the same degree. This difference in optimization is due to the significantly higher FLOPs requirement of standard convolution. Although the L3 access rate is optimized, the main limitation becomes the computational complexity of standard convolution as well as the microprocessor speed.

The Intel Core i5-7600K processor features a 4 MB shared Last Level Cache (LLC), which acts as an L3 cache for the CPU and the primary cache for the integrated iGPU. Thus, Figure 3 also includes measurements of L3 cache missed events for the integrated GPU as well. Similar to Section 2.1, networks were quantized to 16-bit floating point precision before deploying to the integrated GPU. We note that when most design patterns are executed on the integrated GPU in FP16, the LLC access efficacy is not as beneficial as it is on the CPU using FP32. Despite this, the GPU-specific accelerated models still exhibit lower network latency when compared to the microprocessor-specific accelerated variants.

Figure 4 examines the impact of increasing the architectural width on network latency. In contrast to increasing architectural depth, network latency correlates well with the number of FLOPS when width is varied. However, we can still note improvement in network latency when the L3 cache access is optimized. Again, in particular, depthwise bottleneck convolution and depthwise separable convolution exhibit the highest speedups. Since increasing the number of output channels roughly quadruples the number of FLOPS, the network latency is quickly bounded by the computational complexity rather than the memory.

## 2.3 The Impact of Hardware-specific Acceleration on Handcrafted & NAS-derived Architectures

As the last experiment, we examine the degree of inference speed gains from hardware-specific acceleration across 12 hand-crafted deep neural networks and NAS-derived deep neural network architectures. The evaluated architectures can be grouped based on core macro-architecture design pattern they leverage including: **i)** standard convolution (ResNet18, ResNet34), **ii)** bottleneck convolution (ResNet50, ResNet101, ResNet152, DARTS ImageNet, Optimized DARTS ImageNet), **iii)** grouped convolution (ResNext50, ResNext101), **iv)** depthwise bottleneck convolution (MobileNetv2), **v)** bottleneck depthwise convolution and factorized convolution (Inceptionv3), and **vi)** NAS design (NasNet-A (large)). Figure 5 reports the network inference times for both native framework and hardware-specific acceleration.

As seen in Figure 5, the inference time improvements is at least 200-300% for all tested network architectures when they deployed via hardware-specific acceleration compared to native framework. More specifically, MobileNetv2 achieves the lowest inference times of 2.4 ms, with a speed gain of approximately 700% under hardware-specific acceleration compared to when is deployed via native framework. This improvement can be attributed to the significant speed gains achieved when leveraging a depthwise bottleneck convolution design pattern as observed in the first experiment explained Section 2.1. The network architecture with the greatest speed gain from

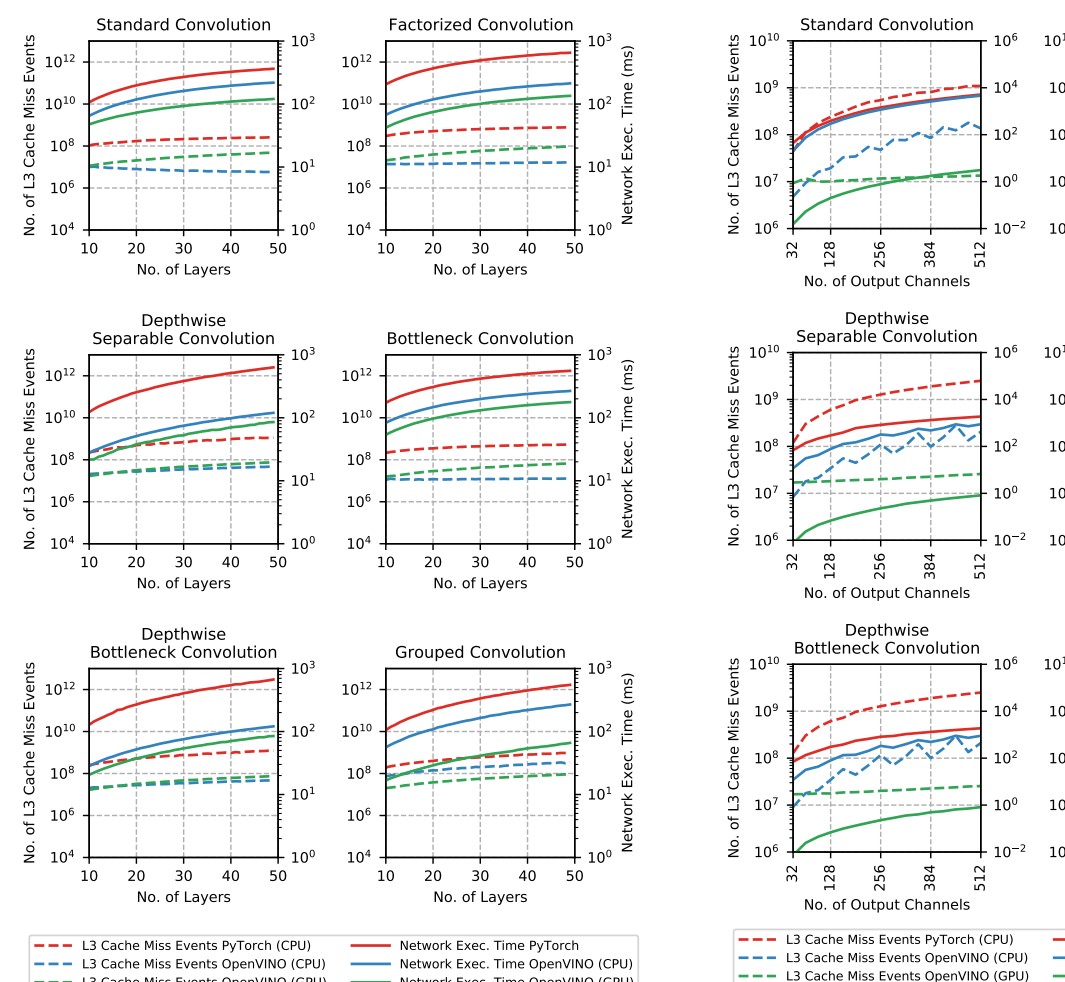

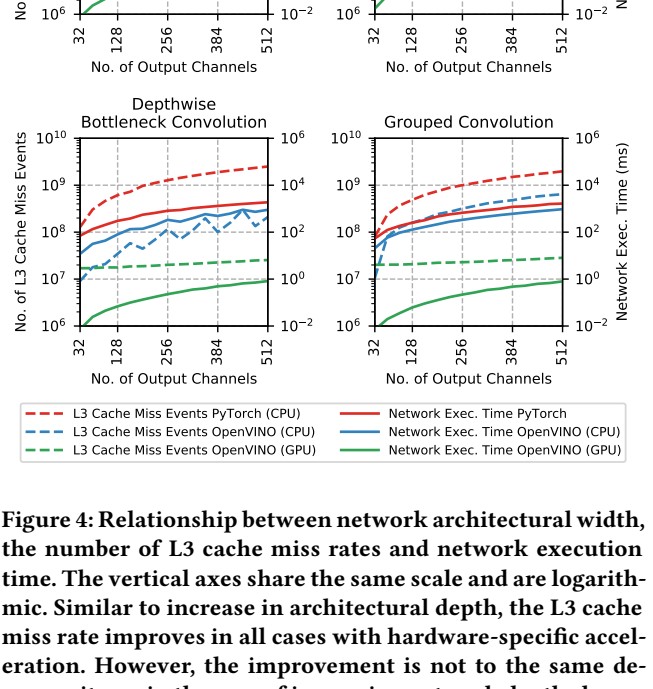

Figure 3: Relationship between network architectural depth, the number of L3 cache miss rates, and network execution speed. The vertical axes share the same scale and are logarithmic. The hardware-specific acceleration reduces the amount of the number of L3 cache misses significantly in all cases. Improving the L3 cache efficacy improves the network execution speed to a varying degree. Depthwise bottleneck convolution benefits the most in terms of execution speed as the network execution improves by about 520%.

Figure 4: Relationship between network architectural width, the number of L3 cache miss rates and network execution time. The vertical axes share the same scale and are logarithmic. Similar to increase in architectural depth, the L3 cache miss rate improves in all cases with hardware-specific acceleration. However, the improvement is not to the same degree as it was in the case of increasing network depth shown in Figure 3. This is because, in case of increasing convolutional channels, the computation is quickly limited by the number of FLOPS.

hardware-specific acceleration is the DARTS-derived network architecture, which saw an improvement of around 1200%. This large speed gain can be attributed to two main reasons. First, DARTS uses separable convolutions exclusively which see significantly greater hardware-specific acceleration when compared to standard convolution (see Figure 1). Secondly, DARTS also employs dilated convolutions in its operations space. However, presently native frameworks such as PyTorch do not optimally execute dilated convolutions. PyTorch only takes advantage of Intel's oneAPI Deep Neural Network library (oneDNN) [6] for non-dilated convolutions

only. Thus, when the network undergoes hardware-specific acceleration (i.e., OpenVINO), the performance is improved significantly. By modifying the PyTorch model to explicitly use oneDNN for all supported operations, the performance difference reduces to about 500%. Nevertheless, the modified model still has significant overheads in terms of changing channel order. Specifically, sometimes PyTorch will change the native layout to match what the math library prefers depending on the operation and operands. Some popular formats are NCHW (channels-first) and NHWC (channels-last) where $N$ is the batch size, $H$ and $W$ are the height and width,

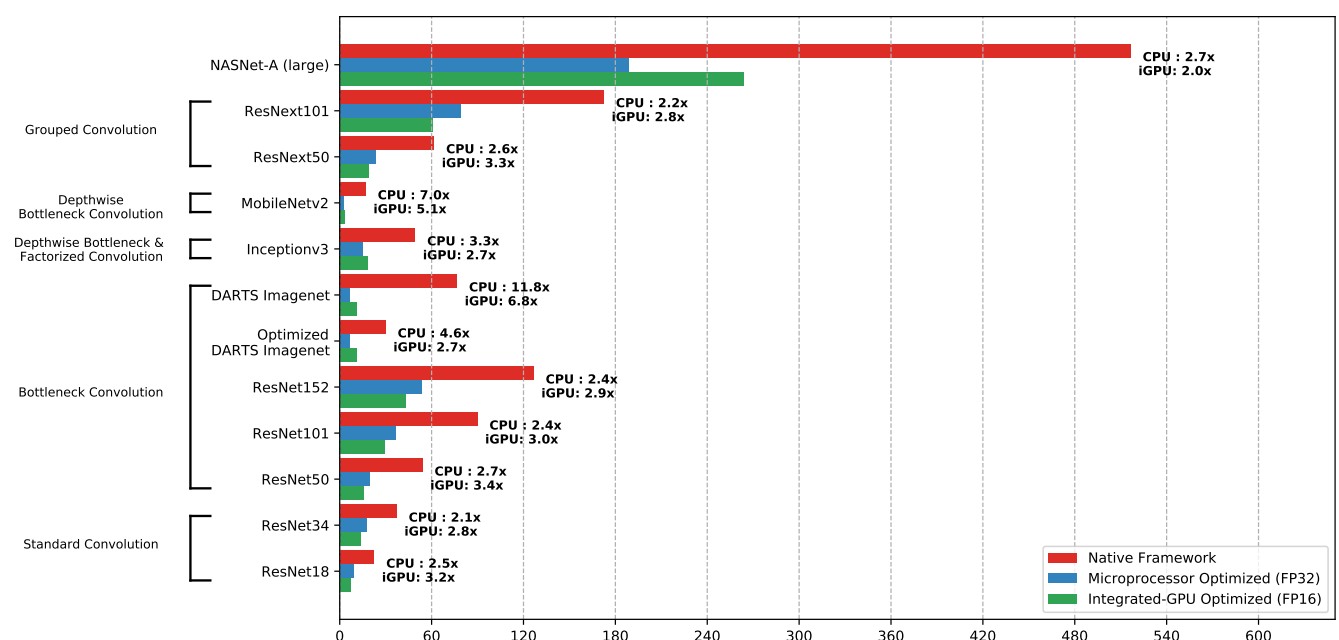

**Figure 5: Inference speed of handcrafted networks and those found via NAS-based strategies under the native framework and hardware-specific acceleration. The bold text adjacent to the bars is the improvement of the inference speed for the corresponding model. All networks were tested with PyTorch. MobileNetv2 model achieved the lowest inference time under hardware-specific acceleration (2.4 ms) due to its use of the depthwise bottleneck convolution design pattern. The DARTS derived architecture benefits the greatest from hardware-specific acceleration with an improvement of about 1200% over the native framework. This is partly due to use of dilated convolution and also due to its use of bottleneck convolution which benefits greatly from hardware-acceleration.**

and $C$ is number of channels. Thus, as the data flows between optimized oneDNN operators and PyTorch operators, the channel ordering is changed automatically by PyTorch at the cost of overhead to satisfy the underlying math library. OpenVINO does not suffer from memory layout overheads as the toolkit transforms the input to the most efficient layout for the operation.

It is worth noting that the experiments in this study were performed in 32-bit floating point precision. As such, network quantization down to 8-bit integers may further enhance the inference speedup provided by OpenVINO. Finally, it can be observed that the degree of speed gain from hardware-specific acceleration varies significantly across different examined architectures, but correlates well with the observations seen in the first experiment based on their core design pattern.

The findings in the conducted experiments illustrate that the choice of architectural design can have a high significant impact on hardware-specific acceleration and thus an essential consideration when designing network architectures tailored to edge deployment. More specifically, based on the findings across the two experiments in this study, the use of depthwise bottleneck convolutions when designing efficient deep neural networks results in the greatest speed gains from the examined hardware-specific acceleration in microprocessor-based deployments, and enables one to build deeper

neural networks to achieve a stronger balance between modeling accuracy and efficiency.

## 3 CONCLUSION

The efficient design and acceleration of deep neural networks are key for widespread deployment of deep learning applications. We explored empirically the fine-grained relationship between form and function from the perspective of architecture design and hardware-specific acceleration. The inference speed of a large number of deep neural network configurations with six different macro-architecture design patterns ranging from 10 to 50 layers depth as well as convolutional widths ranging from 32 to 512 were studied. The experimental results showed that the degree of inference speed gains achieved with hardware-specific acceleration can vary greatly depending on the macro-architecture design pattern choice. However, depthwise bottleneck convolution and depthwise separable convolution design patterns achieve the greatest degrees of acceleration especially at greater architecture depths. Additionally, the relationship between L3 cache optimization, network latency, FLOPs requirement, and architectural depth and width was investigated. It was noted that to take full advantage of computationally inexpensive design patterns, such as depthwise bottleneck convolution, it was necessary to optimize the L3 cache access such that the CPU does not have to fetch data from the DRAM. Finally, by further

studying 12 different hand-crafted and architecture-searched deep neural network under hardware-specific acceleration as well as the native framework, we found that inference speed gains also vary greatly depending on the overall network architecture. The results showed that the speed gain correlates well with the design patterns leveraged in their overall architecture design, as well as the use of operations (e.g., dilated convolutions) that are greatly accelerated via hardware-specific acceleration. One limitation of this study is that the empirical exploration focuses on microprocessor (FP32) and integrated-GPU (FP16) specific acceleration, and thus future work involves studying the impact of architecture design on hardware-specific acceleration for different hardware designs such as INT8 and BFloat16 [16] acceleration on CPUs, variable precision on FPGAs and other hardware acceleration on discrete GPUs. Furthermore, we aim to incorporate such insights into the construct of neural architecture search to enable more efficient identification of deep neural network architectures with great balance between accuracy and real-world efficiency.

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
