# OpenReview forum: "Does Form Follow Function? An Empirical Exploration of the Impact of Deep Neural Network Architecture Design on Hardware-Specific Acceleration"
_tinyml.org/tinyML/2021/Research_Symposium — tinyML 2021 Poster_

### Official Review · AnonReviewer1 · 2021-01-29

**Overall Merit Score:** 1

**Brief Summary:**

This paper compares execution time and L3 cache misses for several neural network layer types and neural networks that are used in Computer Vision. These comparisons are carried out on an early 2017 Intel processor, the Core i5-7600K, that was targeted at low to midrange desktop PC platforms. Results are compared for the following configurations:

- “Native” framework: PyTorch running on the CPU.

- “Hardware Accelerated”: OpenVINO framework running on the CPU, and OpenVINO running on the integrated Intel HD 630 GPU.
Network depth, batch size and layer width are varied as well.

The authors conclude that depthwise convolutions achieve the greatest degree of “hardware” acceleration (PyTorch CPU vs. OpenVINO CPU), as do dilated convolutions. They also point out that OpenVINO achieves a lower L3 cache miss rate than PyTorch.


**Detailed Comments:**

- The paper lumps both CPU and GPU results into the “Hardware accelerated” category, which encompasses OpenVINO on the CPU as well as on the integrated GPU. This is compared against a “Native” baseline of PyTorch on CPU. The result is that it is difficult for the reader to separate the impact of a more efficient SW framework from the impact of specialized HW.

- The paper uses a low-end platform that was designed in the days before NN workloads were significant for low-end desktop PCs. As a result, it is difficult for the reader to translate the findings of the paper to contemporary CPU or GPU systems.

- Results relating to speed (Fig. 1 and 2 for instance) are presented as a single “Network execution time” value, instead of the more traditional latency plus throughput measurements. The single number makes it more difficult to extrapolate the results to specific applications that may be sensitive to latency, throughput or both.

- Cache miss data in Fig. 3 is presented in absolute cache miss events, which is difficult for the reader to interpret. It would be more helpful to present the data using relative metrics such as MPKI (cache misses per K instructions), or misses relative to activation/weights, or misses per FLOP, etc.

- The paper offers little by the way of explanation or analysis for the results they present. For instance, the data from Fig. 1 shows that OpenVINO is much faster than PyTorch on CPU, and that for certain layer types (depthwise convolutions), OpenVINO on GPU is not much faster than OpenVINO on CPU. The underlying reasons for these findings are only very briefly discussed and would merit further analysis in terms of SW framework efficiency (operator fusion, underlying libraries, etc), layer characteristics (arithmetic intensity, activation/weights size), and data transfer schedule [9] for weights and activations.

- Similarly, per Fig. 3, OpenVINO on CPU has far lower LLC miss rates than PyTorch on CPU; it would be of interest to understand how this is achieved.

- In order to help the reader better understand how efficient the SW frameworks are, it would be helpful to include summary data on the characteristics of the CPU and GPU, such as cache hierarchy for the CPU, buffer memory hierarchy for the GPU, and theoretical lower bounds on achievable cache miss rates for optimal data transfer schedules [9] for each NN layer type studied. It would also be useful to know if “Turbo” mode on the CPU was on or off, as it can significantly affect measurements.

- Re. the insights on lines 315-326, it is well known that depthwise convolutions perform well on CPUs. Also, the growth pattern described for standard convolutions is as expected from the growth in FLOPs, and well known. Similarly, the pattern described in lines 294-303 is well known—however, it would be of interest to explain it in terms of arithmetic intensity and available HW parallelism.

- The conclusion in lines 570-579 of the paper is erroneous. Fig. 5 shows that MobileNetV2 runs 7x faster on OpenVINO CPU vs. PyTorch CPU; the authors view this as a “hardware specific acceleration” (lines 574-575), while the same figure shows that MobileNetV2 runs somewhat slower on OpenVINO GPU (5.1x) compared to OpenVINO CPU (7.0x). It is well known that the MobileNet family of NNs [2] has been optimized to run on CPU—specifically mobile CPUs—hence their GPU performance potential is limited. By lumping OpenVINO CPU SW acceleration into the “hardware acceleration” category, the authors erroneously conclude that MobileNets accelerate well in HW.

References

[1] Wu, Bichen, Xiaoliang Dai, Peizhao Zhang, Yanghan Wang, Fei Sun, Yiming Wu, Yuandong Tian, Peter Vajda, Yangqing Jia, and Kurt Keutzer. "Fbnet: Hardware-aware efficient convnet design via differentiable neural architecture search." In Proceedings of the IEEE/CVF Conference on Computer Vision and Pattern Recognition, pp. 10734-10742. 2019.

[2] Howard, Andrew G., Menglong Zhu, Bo Chen, Dmitry Kalenichenko, Weijun Wang, Tobias Weyand, Marco Andreetto, and Hartwig Adam. "Mobilenets: Efficient convolutional neural networks for mobile vision applications." arXiv preprint arXiv:1704.04861 (2017).

[3] Deng, Lei, Guoqi Li, Song Han, Luping Shi, and Yuan Xie. "Model compression and hardware acceleration for neural networks: A comprehensive survey." Proceedings of the IEEE 108, no. 4 (2020): 485-532.

[4] Reuther, Albert, Peter Michaleas, Michael Jones, Vijay Gadepally, Siddharth Samsi, and Jeremy Kepner. "Survey and benchmarking of machine learning accelerators." In 2019 IEEE high performance extreme computing conference (HPEC), pp. 1-9. IEEE, 2019.

[5] https://inst.eecs.berkeley.edu/~ee290-2/sp21/

[6] Sze, Vivienne, Yu-Hsin Chen, Tien-Ju Yang, and Joel S. Emer. "Efficient processing of deep neural networks." Synthesis Lectures on Computer Architecture 15, no. 2 (2020): 1-341.

[7] Yang, Yifan, Qijing Huang, Bichen Wu, Tianjun Zhang, Liang Ma, Giulio Gambardella, Michaela Blott et al. "Synetgy: Algorithm-hardware co-design for convnet accelerators on embedded fpgas." In Proceedings of the 2019 ACM/SIGDA International Symposium on Field-Programmable Gate Arrays, pp. 23-32. 2019.

[8] https://neuralmagic.com/

[9] Parashar, Angshuman, Priyanka Raina, Yakun Sophia Shao, Yu-Hsin Chen, Victor A. Ying, Anurag Mukkara, Rangharajan Venkatesan, Brucek Khailany, Stephen W. Keckler, and Joel Emer. "Timeloop: A systematic approach to dnn accelerator evaluation." In 2019 IEEE international symposium on performance analysis of systems and software (ISPASS), pp. 304-315. IEEE, 2019.

**Paper Strengths:**

- This paper covers a well-studied topic in a mature field with a wide body of prior work. A small sampling of such work can be found in the References section below. Some of the known techniques include co-optimizing neural network design and implementation on CPU [1][2] and co-optimizing neural network design and FPGA implementation [7]. In addition, there are surveys [3][4], classes [5], and books [6] on this topic. There are also commercial efforts focused on optimizing NNs on CPUs, such as [8].

- The paper makes two interesting contributions. The paper shows that OpenVINO runs several times faster than PyTorch on the CPU, and also achieves far lower LLC miss rates. The authors briefly describe possible explanations in lines 327-240. Another interesting finding is that OpenVINO sometimes runs faster on the CPU than on the integrated GPU. The underlying reasons for this are not discussed.





**Paper Weaknesses:**

- The results are not particularly novel or unexpected given the maturity of the field and are based on an obsolete platform that lacks any features for DNNs.
- Lack of any attempt at analysis to explain the results, for instance based on arithmetic intensity, layer characteristics, data transfer schedule, buffer memory or cache structure.
- The paper lumps CPU and GPU results into the “hardware accelerated” category, which makes interpretation of SW vs. HW impact difficult.
- Results are presented in non-standard ways, further adding to the lack of clarity.
- Erroneous conclusion regarding the suitability of MobileNets for HW accelerators.


**Poster (If Paper Is Rejected):**

1: No, paper is below bar for poster as well

**Reviewer Confidence:**

5: The reviewer is absolutely certain that the evaluation is correct and very familiar with the relevant literature

---

### Official Review · AnonReviewer3 · 2021-01-29

**Overall Merit Score:** 2

**Brief Summary:**

This paper performs an empirical study to measure the speedup of different classes of deep networks on Intel Core i5-7600K CPU’s SIMD and Intel HD Graphics 630 GPU’s SIMT architecture against single threaded runs. It characterizes the sensitivity of different networks parameters (number of channels, network depth, etc.) on network speedup, and other hardware configurations such as L3 cache miss events, etc. It also reports speedup of a few NAS-derived networks on the above-mentioned Intel architecture.

**Detailed Comments:**

In its current form, the novelty of the empirical studies reported in this work is minimal; there have been significant prior literature in this area. Furthermore, as this characterization is primarily done on specific hardware model with OpenVINO like specific deep learning optimization toolkit, it is difficult to generalize for other hardware architectures/deep learning platforms. While different networks are expected to produce different latency and speedup (Figure 5), the paper does not make a strong case for considering one network over others for deployment on edge devices as mentioned in section 2.3 (line 743-747). Pareto curve (Accuracy-latency/size) commonly decides the efficacy of one network over others. In addition, I am not sure how the latency-only characterization of six macro-architecture design patterns can drive their use for designing future network architectures tailored to edge devices.
Minor comments:
-Typo in line 20:  greaxft -> great
-Overall writing quality could be improved


**Paper Strengths:**

Detailed measurements of speedup using different neural network architectures (networks with depthwise, standard, grouped, factorized convolutions, etc.) on commodity CPU and GPU processors

**Paper Weaknesses:**

Low novelty, empirical observations and conclusions in this work are well studied, hardware characterization of different types of networks have been done in the past.

No comparison against prior works. There has been significant research on characterizing deep networks on various hardware platforms with different deep learning compiler and optimization frameworks; https://arxiv.org/pdf/1909.04783.pdf is just one of them.

Results reported only for 32 and 16-bit network architectures; no results for commonly used 8-bit quantized versions of the well-known networks architectures.


**Poster (If Paper Is Rejected):**

1: Yes, ok for poster sesion to nurture work

**Reviewer Confidence:**

4: The reviewer is confident but not absolutely certain that the evaluation is correct

---

### Official Review · AnonReviewer4 · 2021-01-30

**Overall Merit Score:** 3

**Brief Summary:**

In the paper, the authors conducted an empirical exploration on the impact of DNN architecture design on the degree of inference speedup that can be achieved via hardware-specific acceleration. A variety of commonly used NN design patterns with different network depths and widths are evaluated with the OpenVINO microprocessor-specific and GPU-specific acceleration. The authors also study the correlation between FLOPs requirements, level 3 cache efficacy, and network latency. Inference time reduction using hardware-specific acceleration for a variety of NN architecture designs is also presented.

**Detailed Comments:**

In the paper, the authors provide an empirical study of the impact of DNN architecture design, including both the block design, network depths/widths, and network architecture, on the degree of inference speedup achieved by the acceleration framework.

The study is interesting and the results are comprehensive. It would be very helpful if the authors can further add more description of the OpenVINO framework to explain the major optimization steps and also, provide more in-depth analysis of the empirical results, e.g., the impact of batch size on per-image inference speed.

**Paper Strengths:**

The paper has the following strengths:
•	The paper provides comprehensive study to compare the inference speedup of OpenVINO on CPU and GPU for different NN design patterns and NN architectures.
•	The study of the correlation between level 3 cache efficacy and network latency provides informative explanation on the speedup.

**Paper Weaknesses:**

The weaknesses of the paper mainly come from the following aspects:
•	Lack of explanation to the OpenVINO framework: it would be very helpful to introduce the major optimizations provided by OpenVINO and why OpenVINO, from the framework aspect, can achieve such dramatic improvement compared to baseline Pytorch.
•	The experimental setup is not clear enough: the authors need to provide detailed information on what the setup for baseline Pytorch and for OpenVINO is, what are the optimizations considered in OpenVINO, as well as how these optimizations are supposed to benefit the hardware.
•	The impact of batch size on the per-image inference speed is interesting but not well explained. More in-depth explanation would be preferred.


**Poster (If Paper Is Rejected):**

1: Yes, ok for poster sesion to nurture work

**Reviewer Confidence:**

4: The reviewer is confident but not absolutely certain that the evaluation is correct

---

### Official Review · AnonReviewer2 · 2021-01-30

**Overall Merit Score:** 3

**Brief Summary:**

The authors investigate the effects of different parameters of neural network model architecture on latency and memory access using different processors and different software frameworks.


**Detailed Comments:**


* It would be helpful to have a dedicated GPU in this comparison, since that one of the most commonly used processors for DNN inference and training.

* Section 2.1 - It would be helpful to show an illustration and/or provide a brief explanation of what the different patterns (depthwise bottleneck convolution, etc.) are, in case a reader is not familiar with all of them.

* Section 2.1 - How are the different design patterns normalized relative to each other?  For example, a standard convolution layer is generally more expressive than a factorized convolution, which would suggest normalizing to accuracy on some benchmark, but that would vary by benchmark.  If the comparisons just involve the same number of feature maps per layer and the same kernel size, please make that explicit.  Related, for the factorized and DW separable convolutions, is there an activation in between the two constituent sub-layers or is it just a linear factorization?

* Section 2 - Are the models just built with random weights or are they trained to anything?  Is there any sparsity-exploitation in the accelerators or anything else that would suggest data-dependency?

* Please connect the data in Figures 1 and two by indicating what channel width was used for Figure 1 and how many layers were used for Figure 2.

* It looks like the difference between PyTorch+CPU adn OpenVino+CPU is much larger in Fig. 1 than in Fig. 2.  For example, depthwise-separable convolution is at least 4x slower on PyTorch in Figure 1, but about 2x slower in Figure 2.  Can you explain this?

* Section 2.2 - Can you briefly describe the methodology for measuring cache misses?

* There is a lot of data here, but it is hard to find the main insights, beyond the notion that HW-specific software (OpenVino) is faster than naive software (PyTorch) and the benefits vary across model architectures.  Some more discussion about how ML engineers can use this information to guide model selection would make this a stronger paper.

* Hooker's paper "The Hardware Lottery" also discussed how available hardware affects software directions.  It would be a relevant reference to cite.

* This paper comes out pretty favorably for OpenVino. (To be clear, there is nothing to suggest that the study was biased towards OpenVino.) If the authors have an affiliation with or funding from Intel, please disclose that in the final paper.



**Paper Strengths:**

* Very detailed data provides a lot for the reader to learn.
* The paper is very systematically organized.
* The paper addresses a very relevant problem in that the easily calculated metrics of model complexity (e.g. # MACs) do not always reliably predict inference speed.


**Paper Weaknesses:**

* While there is a great deal of data, it is hard to discern the key insights.  From the standpoint of an ML engineer, it is not clear what to actually *do* with this information (other than maybe get OpenVino?).
* The set of hardware targets is rather limited.  Inclusion of a standalone GPU would improve the relevance.


**Poster (If Paper Is Rejected):**

1: Yes, ok for poster sesion to nurture work

**Reviewer Confidence:**

4: The reviewer is confident but not absolutely certain that the evaluation is correct

---

### Decision · Program_Chairs · 2021-02-05

**Decision:**

Accept (Poster)

**Comment:**

Based on the reviewer feedback, your paper has been accepted as a poster.

Please read the reviews carefully and make sure the concerns are addressed in your poster submission.

Accepted posters are given a 5-minute slot for an oral presentation on Friday, March 26, 2021, to pitch the main ideas of your work and to stimulate discussions. Detailed instructions will follow soon. All final posters will earn a stamp of acceptance as such: “Published as a poster at TinyML Research Symposium 2021.”